# Evaluating the Transition from Targeted to Exome Sequencing: A Guide for Clinical Laboratories

**DOI:** 10.3390/ijms24087330

**Published:** 2023-04-15

**Authors:** Kevin Yauy, Charles Van Goethem, Henri Pégeot, David Baux, Thomas Guignard, Corinne Thèze, Olivier Ardouin, Anne-Françoise Roux, Michel Koenig, Anne Bergougnoux, Mireille Cossée

**Affiliations:** 1Laboratoire de Génétique Moléculaire, LGM, Centre Hospitalier Universitaire de Montpellier, IURC—Institut Universitaire de Recherche Clinique, 641 Avenue du Doyen G. Giraud, 34090 Montpellier, Francec-vangoethem@chu-montpellier.fr (C.V.G.);; 2Service de Génétique Médicale, CHU Montpellier, 371 Avenue du Doyen G. Giraud, 34090 Montpellier, France; 3INM, Université de Montpellier, INSERM, Hôpital Saint Eloi-Bâtiment INM 80, rue Augustin Fliche-BP 74103, 34090 Montpellier, France; 4Unité de Génétique Chromosomique, Département de Génétique Médicale, Maladies Rares et Médecine Personnalisée, Hôpital Arnaud de Villeneuve, CHU de Montpellier, 371 Av. du Doyen Gaston Giraud, 34090 Montpellier, France; 5Plateau de Médecine Moléculaire et Génomique, Hôpital Arnaud de Villeneuve, CHU de Montpellier, 34090 Montpellier, France; 6PhyMedExp-Physiologie et Médecine Expérimentale du Cœur et des Muscles, Université de Montpellier, Inserm U1046, CNRS UMR 9214, 371 Avenue du Doyen G. Giraud, 34090 Montpellier, France

**Keywords:** NGS, exome sequencing, targeted sequencing, quality metrics, diagnostics

## Abstract

The transition from targeted to exome or genome sequencing in clinical contexts requires quality standards, such as targeted sequencing, in order to be fully adopted. However, no clear recommendations or methodology have emerged for evaluating this technological evolution. We developed a structured method based on four run-specific sequencing metrics and seven sample-specific sequencing metrics for evaluating the performance of exome sequencing strategies to replace targeted strategies. The indicators include quality metrics and coverage performance on gene panels and OMIM morbid genes. We applied this general strategy to three different exome kits and compared them with a myopathy-targeted sequencing method. After having achieved 80 million reads, all-tested exome kits generated data suitable for clinical diagnosis. However, significant differences in the coverage and PCR duplicates were observed between the kits. These are two main criteria to consider for the initial implementation with high-quality assurance. This study aims to assist molecular diagnostic laboratories in adopting and evaluating exome sequencing kits in a diagnostic context compared to the strategy used previously. A similar strategy could be used to implement whole-genome sequencing for diagnostic purposes.

## 1. Introduction

Next-generation sequencing (NGS) technology is routinely used by clinical diagnostic laboratories to identify variants and genes underlying human genetic diseases. Multiple strategies based on gene-panel sequencing (GPS), exome sequencing (ES), or genome sequencing (GS) exist, according to the clinical situation [1,2]. GPS represents the cheapest and fastest NGS approach [3], with a deeper coverage important for the detection of mosaicism [4]. However, GPS requires frequent updates considering new disease-causing genes. The use of ES in clinical practice has reported evidence of better cost-effectiveness and clinical utility in various indications [5,6,7]. GS is the most powerful, allowing the detection of structural variants as well as deep intronic mutations that may affect splicing. However, because of its cost, time, and the workflow infrastructure requirements, GS is currently mainly accessible to nationwide-scaled projects or in large-scale clinical genomic sequencing centers [8,9].

In the late 2000s and early 2010s, most of the genetic diagnostic laboratories started to apply the GPS strategy to replace Sanger sequencing [10]. This approach has been reported to be efficient and widely adopted in clinical sequencing centers [3,11]. Guidelines for genetic diagnostic laboratories with reliable and accurate evaluation to apply the ES strategy instead of GPS is essential for their operation.

Our diagnostic laboratory previously implemented a GPS strategy for diagnosing myopathies and muscular dystrophies, especially for the giant titin and nebulin genes [3]. To develop an ES strategy with at least the same reliability as GPS, we report here a complete methodology of comparison based on several guidelines previously reported, structured into four main sections: theoretical evaluation of coverage, sequencing quality validation, clinical validation, and final selection of a strategy. We illustrate this method with a comparative study of NGS results obtained on 27 DNA samples using three different exome capture kits.

## 2. Results

We have defined a general strategy with four main steps to evaluate the performance of ES solutions compared to GPS. The general workflow is described in Figure 1.

We used this general strategy to compare ES data from three library preparation solutions that differ in the genomic captured regions concerning the introns and untranslated regions (UTR), the boosted capture in disease-associated regions, and the capture technology: (i) SeqCap EZ MedExome (Medex) from Roche (Santa Clara, CA, USA); (ii) SureSelect Human All Exon v7 (SSV7) from Agilent Technologies (Santa Clara, CA, USA); and (iii) SureSelect Clinical Research Exome V2 (CREV2) from Agilent Technologies (Santa Clara, CA, USA). For each exome capture kit, nine distinct DNA samples were extracted from blood and fragmented according to the same protocol, then sequenced using an Illumina NextSeq500 as paired-end 2 × 150 bp reads (see Materials and Methods).

### 2.1. Theoretical Evaluation of Regions of Interest Coverage

The different exome sequencing kits have specific characteristics regarding the genomic sequences captured and the possible enrichment of disease-associated genes.

To choose the appropriate kit, one must analyze the regions each kit covers. Roche’s SeqCap EZ MedExome (Medex) offers enhanced exon coverage for medically relevant genes. Agilent Technologies’ SureSelect Human All Exon v7 (SSV7) utilizes RefSeq, GENCODE, CCDS, and UCSC Known Genes to focus on the interpretable genome. Their SureSelect Clinical Research Exome V2 (CREV2) builds upon the Human All Exon V6 design, increasing coverage in disease-associated regions and targeting a larger portion of introns and untranslated regions (UTRs).

The genome regions targeted by the different ES designs were compared to the genome assembly GRCh37 (hg19). As expected, all the exome kits have a very similar coding sequence target and mainly differ concerning the non-coding regions (introns and UTR) targeted (Figure 2). CREV2 has comparatively a very extended non-coding target (32.1 Mb) compared to Medex and SSV7 (12.7 and 15.4 Mb, respectively).

### 2.2. Sequencing Quality Validation

The first step to evaluate each ES kit is to ensure the raw sequencing quality. For this, we propose four criteria with acceptable thresholds represented in Table 1: density of clusters, clusters passing filter, quality score Q30, and PhiX control. The density of clusters refers to the number of clusters on the flow cell. Cluster passing filters are defined as clusters that pass the filter based on various parameters, such as signal intensity and purity. Quality score Q30 refers to the percentage of bases with a quality score of 30 or higher. Finally, PhiX control is a measurement of the sequencing performance by using a known library. In our experiments, NGS with the three kits reached the required quality scores.

Then, we recommend assessing sample sequencing quality with seven parameters, according to NGS practice guidelines [27,31,32]: insert size, PCR duplicate rate, on-target rate, depth of coverage, coverage rate, uniformity of coverage, and Ts/Tv ratio. Table 1 compiles each quality parameter with the tools to assess it, acceptable thresholds, common causes, corrective measures in case of unacceptable criteria, and publications or other sources reported.

The results obtained for each ES kit tested are represented in Table 2.

As expected, due to the same mechanical fragmentation protocol used for the three experiments, produced fragments are of similar size (Medex: 206 bp; SSV7: 215 bp; CREV2: 204 bp). Paired-end 150 bp sequencing produced overlapping coverage of 94 bp, 85 bp, and 96 bp for Medex, SSV7, and CREV2, respectively. The proportion of PCR duplicated reads was one of the major differences between the methods as both Agilent kits displayed ~2-fold less PCR duplicates than the Roche MedExome kit. The on-target rates were similar for both kits from Agilent Technologies (~72%) but were slightly higher for the MedExome (~74%). Each of these technologies achieved a high level of respective target region cumulative coverage, the Agilent SSV7 presented the highest. In order to have at least 90% of targeted bases covered at 30X, a minimum of 80M reads were required for Medex and SSV7, whereas 100M was needed for CREV2. The three technologies showed a satisfactory percentage of targeted bases covered at 15X at a sequencing effort as early as 40M (Medex: 96.3%; SSV7: 97.9%; CREV2: 93.9%) (Figure 3).

To investigate the uniformity of coverage, we have computed two metrics, the fold 80 base penalty and the evenness score [27]. Lower numbers for the 80 base penalty and a high percentage for this score indicate more uniform coverage; a value of 1 for the 80 base penalty and 100% for this score represent perfect uniformity. Both metrics indicated the same order of magnitude for the three capture protocols, with the SSV7 kit having slightly higher performances.

We also evaluated the ratio of transitions to transversions (Ts/Tv) in the dataset as they are an approximate measure of variant calling quality [33]. For human-exome sequencing data, the Ts/Tv ratio is generally around 3.0 and about 2.0 outside the exome regions [30]. Ts/Tv ratios in this value range are associated with lower false positives, with high-quality exome variant datasets expected to have Ts/Tv ratios between 2.8 and 3.0 [34]. Based on this statement, all the methods evaluated were of good quality as SSV7 has a Ts/Tv ratio of 2.6 and Medex of 2.7. CREV2 has a lower value (2.4) that could be explained by the higher proportion of non-coding regions in its design.

### 2.3. Clinical Validation

In addition to a technology comparison focused on metrics, it is essential to validate the different ES kits in the clinical context of genetic diseases. This includes ensuring that panels of genes previously analyzed in the laboratory by GPS for diagnosis (particularly regions that are difficult to sequence), as well as coding sequences of disease-associated genes (OMIM genes database [35]), are well covered using ES.

We measured the coverage provided by the three ES kits on regions targeted by our panel of genes implicated in myopathies, routinely used in our laboratory [3] (Figure 4a). 80M reads were necessary to achieve a 30X coverage on more than 99% of the targeted regions and 50M were required to reach a coverage of 30X on more than 95% of the targeted regions. We also evaluated on Integrative Genomic Viewer (IGV) [36,37] the coverage of the repeated regions of *TTN* (exons 172 to 180, 181 to 189, and 190 to 198) and *NEB* (exons 82 to 89, 90 to 97, and 98 to 105) that were not adequately covered by older ES kits tested in a previous study [3]. The three ES kits achieved a similar coverage to that obtained with GPS (Appendix A).

To evaluate the different ES kits in the context of all genetic diseases, we calculated sequence coverage levels obtained with Medex, SSV7, and CREV2 on the OMIM genes dataset. The percentage of regions covered at different sequencing depths (15X, 30X, 50X, and 100X) showed that a sequencing effort of 80M reads is required to cover 95% of the OMIM set with a coverage of at least 30X for the three ES kits (Figure 4b).

### 2.4. Final Selection of a Strategy

In summary, all evaluated ES kits met clinical diagnostic quality standards based on four run sequencing metrics and seven sample sequencing metrics. Upon reaching 80 million reads, all three kits effectively covered at least 90% of targeted bases at 30X coverage in our GPS and OMIM gene coding regions. The primary factors influencing our ES strategy selection were the PCR duplicate rate, which varies among library kits, and coverage of clinically relevant regions. Notably, the larger target size in the CREV2 kit may present financial constraints for many clinical laboratories when implementing routine diagnostic sequencing.

## 3. Discussion

In our study, we implemented a strategy to evaluate several ES technologies in order to assess their reliability and adequacy to replace GPS for variant detection in diagnostic use.

First, it is necessary to evaluate the target design philosophy of each manufacturer. Exome kits usually present very similar coding sequence targets and mainly differ concerning the non-coding regions (introns and UTR) targeted. While few solutions are scheduled to harbor very extended non-coding targets (cover more than 30 Mb), many designs focus on coding regions that make them smaller and thus less expensive. These differences underpin various applications: the kits including extended non-coding targets aim to sequence many non-coding regions of interest (including enhancers), whereas the kits limited to coding sequences are easier for data interpretation and storage in the perspective of routine sequencing for diagnostic purposes and provide a higher sequencing depth that could be useful to detect mosaicism. The selection of exome strategy should be done according to clinical regions and molecular mechanisms involved in the explored pathology.

It is then important to use metrics to evaluate NGS sequencing quality. We provide a compilation of metrics usually used, with tools to evaluate them, acceptable threshold(s) if relevant, common causes and corrective measures in case of results outside the thresholds, and available sources. Evaluation of the performance of each ES kit for each in silico gene panel used in clinical diagnostics is also important. In our study, we focused on myopathy genes’ panel analyzed in our diagnostic laboratory and observed results with quality standards for clinical diagnostics. In addition, when investigating visually the coverage on the repeated regions of *TTN* and *NEB*, we observed performances comparable to GPS. This is an actual improvement compared to the previous generation of exome capture solutions that were evaluated in our laboratory [3]. In a broader way, the performance of the ES capture kits on clinical gene regions coverage is also essential to evaluate for different sequencing efforts. It is important to determine the minimum number of reads per patient to sequence > 95% of clinical gene regions defined in the OMIM database. In our example, we showed that the three tested ES kits demonstrate performance suitable to clinical diagnostic quality standards. This clinical validation is essential because it assures us of the correct sequencing of the myopathy genes of our initial panel and of the OMIM genes, which will improve our diagnostic yield of myopathies [3].

A limitation of our study is that the experiment was solely based on blood DNA samples and was mainly focused on detecting constitutional variants. While an individual’s DNA remains consistent, the genetic composition may vary between different tissues. This variation can include differences in genetic variants or structural alterations within the exome, depending on the specific tissue. Although potential discrepancies between blood DNA and tissue DNA exomes may exist, it is worth noting that these differences may not necessarily be significant but could be in a cancer sequencing context [38]. Finally, in this study, we did not explore the performance of ES in additional variant detection in off-target reads, as mitochondrial variants [39].

More than an increased diagnostic yield, merging all GPS into a unique ES technique could lead to easier work sharing between teams, fewer wet lab updates, and, therefore, less work for validation and accreditations. Of course, the increased number of sequence data obtained by ES, compared to GPS, implies higher storage capacities and adapted analysis pipelines. A strategy for reporting results in case of incidental findings should also be decided, according to international and national recommendations [40,41,42,43]. Moreover, GPS trio sequencing does not have a higher diagnostic yield than an ES trio sequencing approach. To justify the additional costs of genome vs. exome sequencing, improvement of structural variation analysis will be required and/or the cost of genome analysis and storage will need to decrease.

In conclusion, our work aims to be a practical guide for molecular diagnostics of genetic disorders, helpful to perform kits’ benchmarking in order to introduce or change ES kits with a high level of quality assurance. A closed strategy could be used to implement GS, which will probably become the upcoming first-tier genetic test in the next years [44].

## 4. Materials and Methods

### 4.1. NGS Experiments

Genomic DNA was extracted from blood samples following the manufacturer’s standard procedure of the FlexiGene DNA kit (Qiagen, Courtaboeuf, France). For all three ES protocols, 100 ng of fragmented genomic DNA with a Bioruptor (Diagenode, Liège, Belgium) was used as input. DNA NGS libraries were prepared according to the manufacturer’s protocol. Final library concentrations were measured with Invitrogen’s Qubit Fluorometer High Sensitivity kit (Carlsbad, CA, USA), and library quality controls were performed on a Bioanalyzer High Sensitivity DNA chip (Agilent Technologies, Santa Clara, CA, USA). Sequencing of each exome capture library was performed using an Illumina NextSeq500 as paired-end 2 × 150 bp reads according to the manufacturer’s protocol (NextSeq System Denature and Dilute Libraries Guide, January 2016). For each technology, nine distinct samples were sequenced (a total of 27 samples) using NextSeq 500/550 High Output Kit v2 cartridge 300 cycles (2 × 150 cycles).

### 4.2. NGS Data Analyses

#### 4.2.1. Data Processing

For data analyses, the nenufaar [45] data analysis pipeline has been used. Briefly, this pipeline performs the secondary analysis from fastq files to BAMs and raw VCFs. It uses, in particular, BWA-MEM [46] for mapping, GATK 3.8 Haplotype Caller [47,48], and Platypus [49] for variant calling. Several quality metrics are generated during the process, such as Picard, Qualimap (BamQC tool) [50,51], GATK CollectHSMetrics, DepthOfCoverage [52], and FastQC [22].

#### 4.2.2. Theoretical Evaluation of Regions of Interest Coverage

The genome regions targeted by the different ES designs were compared to the genome assembly GRCh37 (hg19) with multiIntersectBed from the bedtools suit [53]. The genomic coordinates of the exon coding sequence (CDS), untranslated regions (UTR), and introns were defined according to the National Center for Biotechnology Information (NCBI) Reference Sequence [54].

#### 4.2.3. Sequencing Quality Validation

The data concerning the four quality criteria of the run (density of clusters, cluster passing filter, quality score Q30, and PhiX control) are provided by the sequencer software (i.e., Illumina Sequencing Analysis). Concerning the seven parameters (insert size, PCR duplicate rate, on-target rate, depth of coverage, coverage rate, uniformity of coverage, and transition/transversion (Ts/Tv) ratio) that ensure that the samples meet the quality requirements for analysis, most of their measures can be provided by GATK picard tools [55]. The uniformity of coverage can be computed using the Evenness Score [27] and the Ts/Tv ratio can be measured using bcftools [56]. For each quality parameter, we compiled the corresponding assessment tool, acceptable threshold, common causes of poor results, and suggested corrective measures in case of unacceptable criteria. In addition, we also provided publications or other sources that reported on each parameter, as summarized in Table 2.

#### 4.2.4. Clinical Validation: Coverage of Targeted Regions

To evaluate coverage, fastq files were down sampled randomly to simulate different sequencing efforts. Coverage efficiency was evaluated by calculating cumulative coverage over all intended target bases for different amounts of reads, 40M (millions of reads), 60M, 80M, and 100M using the seqtk package (https://github.com/lh3/seqtk accessed on 28 March 2022).

The genomic locations targeted by different ES kits for at least a given coverage rate were computed using Qualimap [51].

The OMIM set was established based on the genes associated with a clinical phenotype description in the OMIM database [35] (March 2019). The gene symbols were used to generate a bed file from the RefSeq database with a 25 base pairs exon padding in order to include nearby canonical sites impacting mRNA splicing.

## Figures and Tables

**Figure 1 ijms-24-07330-f001:**
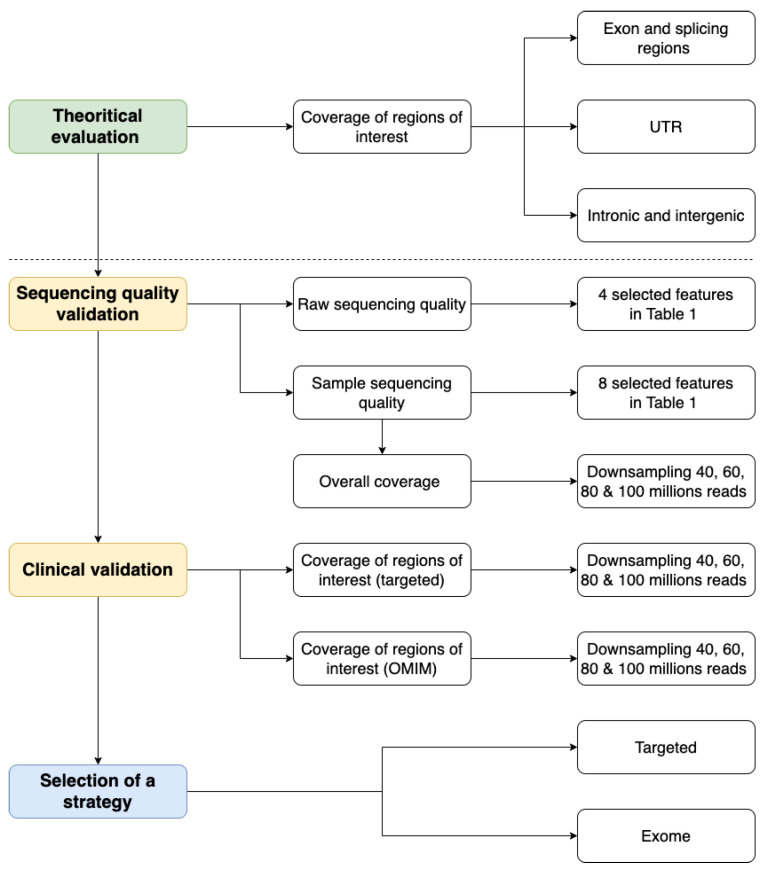
Evaluating the transition from targeted to exome sequencing workflow.

**Figure 2 ijms-24-07330-f002:**
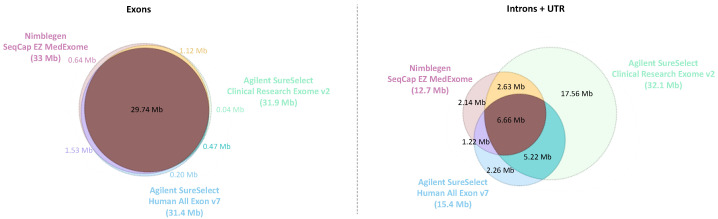
Venn diagram showing the overlap of targeted bases of the different ES designs for non-coding (UTR + intron) regions and coding regions. The MedExome (Medex) has a target size of 47 Mb covering 33 Mb of exonic regions, 10.8 Mb of intronic regions, and 1.9 Mb of UTR. The SureSelect Human All Exon V7 exome (SSV7) presents a target size of 48 Mb composed of 31.9 Mb of exonic regions, 13.4 Mb of intronic regions, and 2 Mb of UTR. The Clinical Research Exome V2 (CREV2) has the largest target size with 67.3 Mb due to its larger intronic regions coverage (25.7 Mb) and UTR (6.4 Mb), whereas the exonic regions size is similar to the two other designs (31.4 Mb). The associated Venn diagrams were generated with Plotly (Plotly Technologies Inc. Collaborative data science. Montréal, QC, Canada, 2015. https://plot.ly accessed on 10 March 2019).

**Figure 3 ijms-24-07330-f003:**
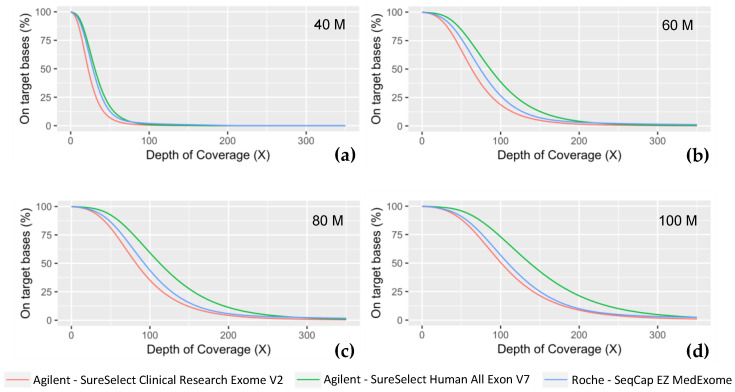
Coverage efficiency obtained with the different ES kits. Coverage efficiency is represented as the percent of the intended targeted bases for each technology at fixed depths (X) for different sequencing effort, simulated with a random down sampling of (**a**) 40 million reads, (**b**) 60M reads, (**c**) 80M reads, and (**d**) 100M reads.

**Figure 4 ijms-24-07330-f004:**
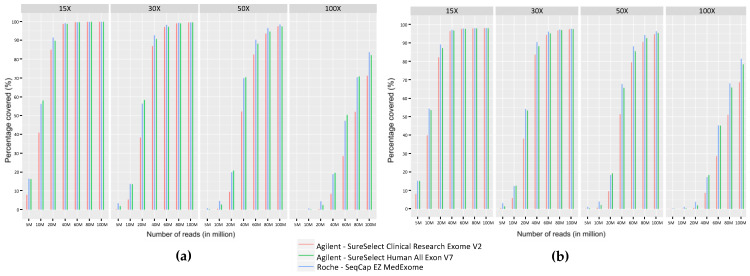
Comparison of coverage as percentage of bases covered at fixed depths (15X, 30X, 50X, and 100X) for different sequencing efforts (5M, 10M, 20M, 40M, 60M, 80M, and 100M) obtained with the three ES kits on (**a**) the panel of neuromuscular genes used in diagnosis in the laboratory [3] and (**b**) OMIM disease-associated genes.

**Table 1 ijms-24-07330-t001:** Sequencing quality validation parameters.

Quality parameters	Description	Tools to Evaluate	Acceptable Threshold(s) (Depending on Context)	Results Outside Thresholds: Common Causes and/or Corrective Measures	Sources
**Raw Sequencing Quality**					[12,13]
Density of clusters (K/mm^2^)	The density of clusters on the flow cell (in thousands per mm^2^). This parameter is a direct representation of the amount of DNA loaded.	Sequencer Software (ex: Sequencing Analysis Viewer from Illumina^®^)	Depends on instruments. For example: **MiniSeq**-High and Mid-170–220 **MiSeq**-v2-1000–1200 **MiSeq**-v3-1200–1400 **NextSeq**-v2 High and Mid-170–220 **HiSeq2500**-v1 and v2-850–1000 **HiSeq2500**-v3-750–850 **HiSeq2500**-v4-950–1050	Inaccurate library quantification is the most common cause of over or under-clustering.	[14,15,16]
Clusters passing filter (%PF)	The %PF is the number of clusters that passed Illumina’s “Chastity filter”. The “Chastity Filter” is a ratio of the brightest base intensity (Ia) divided by the sum of the brightest and second brightest (Ib) base intensities: *Ia*/(*Ia* + *Ib*). A cluster does not pass this filter if 1 base call has a chastity value below 0.6 in the first 25 cycles.		65%	In the most common cases, a %PF under 65% is due to an over-clustering.	[17,18]
Quality score Q30 (%)	The percentage of bases with a phred quality score of 30 or higher. Phred-like quality scores (Q-scores) are used to measure the accuracy of nucleotide identity data from a sequencing run. This value is an average across the whole read length since error rate increases towards the end of the reads. *Q* = −10.*log*10(*e*) Error rate: percentage of bases called incorrectly at any one cycle. Q30 is the best indicator to check base quality.		80%. This threshold may be adapted following DNA quality; if the sample is from FFPE or is old then the DNA may be of poor quality but precious.	The main cause of a low Q30 is the poor quality of DNA. The extraction is a key step. Another cause is the quality of the reagents or polymerase, the reason why the Q30 score decreases as the run progress.	[19,20]
PhiX control (%)	PhiX is an adapter-ligated library used as an internal control for Illumina sequencing run quality monitoring. PhiX% is calculated from the reads that are aligned to Illumina’s PhiX control.		>0.3% Ideally preconized around 1%.	The less complex/diverse is the library, the higher PhiX control amount is needed.	[21]
**Sample Sequencing Data Quality**					
Insert size	Median or mean length of sequenced fragments calculated from fastq.	FastP Picard (GATK) FastQC	Around 200–250 Depending on library kits.	Adjusting fragmentation could lead to an optimal sequencing and coverage uniformity.	
Duplicate rate	Rate of deduplicated reads.	Picard (GATK) FastQC	An acceptable threshold is under 20%. Depending on library kit, targets or depth.	Can be diminished by optimizing the amount of starting material and the number of PCR cycles in the laboratory.	[22,23]
On-target rate	Percent of sequencing data/reads which maps to regions of interest: ratio of the number of sequenced bases covering the target regions to the total number of mapped bases output by the sequencer.	Picard (GATK)	An acceptable threshold is >80%. Depending on library kit, targets or depth.	Substantially influenced by insert size.	
Depth of coverage	Median or mean coverage on all target bases (expressed in X).		Strongly recommended, at least, 100X. Depending on application.	For a better uniformity of coverage, a lower threshold is acceptable. Lower numbers of samples will increase coverage.	[24]
Coverage rate (% at nX)	Percent of target bases with coverage > nX.		Strongly recommended: >90% at 30X. Depending on application, targets, or library.	Lower numbers of samples will increase coverage. A change in capture design or technology should increase the coverage rate.	
Uniformity of coverage	Homogeneity in coverage of the NGS targets, represented by the evenness score (ES) and fold 80 base penalty (Fold-80). The fold 80 base penalty is defined as the fold change of non-zero read coverage needed to bring 80% of the targeted bases to the observed mean coverage.	MiSeqReporter/Local Run Manager HomeMade Script	Threshold depending of the method of calculation. A lower value of the Fold-80 and a high percentage of the ES indicate less variability among the coverage of the individual targets, a value of 1 of the Fold-80 base penalty, and of 100% of the ES representing a perfect uniformity.	A change in capture design or technology should increase the coverage rate.	[25,26,27]
Ts/Tv ratio (SNV)	Transitions (Ts) (changes from A <-> G and C <-> T) compared to transversions (Tv) (changes from A <-> C, A <-> T, G <-> C or G <-> T)	BCFTools SNPSift *GATK* *VariantEval* (*BETA*)	An acceptable threshold on CDS sequencing is >2.4. Depending on the application.	Across the entire genome, the ratio of transitions to transversions is typically around 2. In protein coding regions, this ratio is typically higher, often a little above 3. This metric can be used as a long-term control, if this metric changes drastically it can mean a problem with the capture, samples, or sequencer.	[28,29,30]

**Table 2 ijms-24-07330-t002:** Sample sequencing data quality obtained for the three ES kits. The best performances are shown in bold.

Number of Reads (Million)	Exome	Median Insert Size (bp)	On-Target Rate	On-Target Mean Coverage with Duplicates (X)	Duplicate Reads (%)	On-Target Mean Coverage without Duplicates (X)	Target Base at 30 X (%)	Fold 80 Base Penalty	Evenness	Ts/Tv Ratio
40M	Medexome	206	**74.26**	61.3	12.09	42.1	66	1.9	77.25	2.8
	SSV7	215	72.04	**66.1**	5.16	**45.5**	**73**	1.8	**79.46**	2.7
	CREV2	204	72.14	48.8	**4.26**	33.3	51	2.0	77.62	2.5
60M	Medexome	206	**74.26**	86.4	17.25	59.5	83	1.9	77.03	2.7
	SSV7	218	72.04	**96.32**	7.54	**66.3**	**89**	1.7	**79.52**	2.6
	CREV2	205	72.14	71.4	**6.24**	48.8	75	1.9	77.99	2.4
80M	Medexome	207	**74.26**	108.4	21.93	74.7	90	1.9	77.22	2.7
	SSV7	218	72.04	**124.9**	9.79	**86.1**	**94**	1.8	**80.01**	2.6
	CREV2	205	72.14	93.0	**8.15**	63.6	86	1.8	78.63	2.4
100M	Medexome	209	**74.26**	127.8	26.19	88.2	92	1.8	77.46	2.7
	SSV7	218	72.04	**151.9**	11.94	**104.9**	**96**	1.7	**80.22**	2.6
	CREV2	206	72.14	113.6	**9.96**	77.7	90	1.8	79.31	2.4

## Data Availability

The data presented in this study are available on request from the corresponding author.

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
