# Peer review of "Evaluating the Transition from Targeted to Exome Sequencing: A Guide for Clinical Laboratories"

_ijms, 2023, doi:10.3390/ijms24087330_

Round 1

Reviewer 1 Report

"Evaluating the transition from targeted to exome sequencing: a guide for clinical laboratories"

1. According to line-58 “18-DNA samples were used for three different exome capture kit”, while line-250 “Total of 27 samples were sequenced” can the author explain and have some clarity?

2. In line-77-86 “2.1: Theoretical Evaluation of regions of interest coverage” This paragraph requires more clarity.

3. The occurrence of Figure 1 and 2 hasn’t mentioned in the main manuscript

4. What is the difference in table-1 and table-1 (continued)? (line 113 and 114)

5. The significance of clinical validation (line 162- 164) can be elaborated.

6. Why coverage of repeated region of was performed only for  TTN and NEB? (line-216) 

Overall Review:

1. The entire experiment is conducted on the blood DNA sample. Though the DNA remains the same for the particular individual the composition between different tissues in the body differs. Difference in mutation, genetic variants or structural variations present in exome depends on the specific tissue. It is equally important to note that any differences between blood DNA and Tissue DNA exomes would likely be same but not necessarily be significant.

Therefore, keeping the context in note does the “Evaluating the transition from targeted to exome sequencing: a guide for clinical laboratories” hold good for tissue DNA samples?

2. The paper writing requires more of dedication as the formatting of table/figure arrangement is not appropriate.

3. The overall paper writing, experiment conducted and creativity is excellent

Author Response

  1. According to line-58 “18-DNA samples were used for three different exome capture kit”, while line-250 “Total of 27 samples were sequenced” can the author explain and have some clarity?

We thank the reviewer for this remark. The 18 number was a copying error, we have effectively sequenced 27 DNA samples. 

Appropriate modifications line-58 : “18 DNA samples” => “27 DNA samples” has been made.

  1. In line-77-86 “2.1: Theoretical Evaluation of regions of interest coverage” This paragraph requires more clarity.

Thank you for this remark, appropriate modifications to improve clarity have been made.

  1. The occurrence of Figure 1 and 2 hasn’t mentioned in the main manuscript

We are confused, we do not understand your comment. Figure 1 is cited at the beginning of the result section in line 61 (originally 63) and Figure 2 in line 88 (originally 91).

  1. What is the difference in table-1 and table-1 (continued)? (line 113 and 114)

Table 1 was duplicated by mistake. As requested by the reviewer 2, appropriate modifications have been made, including sources of the table as references. 

  1. The significance of clinical validation (line 162- 164) can be elaborated.

Thank you for this comment. We have clarified this concept by rearranging the chapter and putting the following text at the beginning of the chapter:

“In addition to a technology comparison focused on metrics, it is essential to validate the different ES kits in the clinical context of genetic diseases. This includes ensuring that panels of genes previously analyzed in the laboratory by GPS for diagnosis (particularly regions that are difficult to sequence), as well as coding sequences of disease-associated genes (OMIM genes database[37]), are well covered using ES.”

  1. Why coverage of repeated region of was performed only for  TTN and NEB? (line-216) 

We are a clinical genetics laboratory specialized in myopathies. As TTN and NEB are known to be difficult genes to sequence and in our domain of expertise, we focused our analyses on these genes to evaluate ES sequencing performance.

Overall Review:

  1. The entire experiment is conducted on the blood DNA sample. Though the DNA remains the same for the particular individual the composition between different tissues in the body differs. Difference in mutation, genetic variants or structural variations present in exome depends on the specific tissue. It is equally important to note that any differences between blood DNA and Tissue DNA exomes would likely be same but not necessarily be significant.

Therefore, keeping the context in note does the “Evaluating the transition from targeted to exome sequencing: a guide for clinical laboratories” hold good for tissue DNA samples?

Thank your for this feedback, we added the followin sentences to highlight this point on the discussion section: 

  “A limitation of our study is that the experiment was solely based on blood DNA samples and mainly focus to detect constitutional variants. While an individual's DNA remains consistent, the genetic composition may vary between different tissues. This variation can include differences in genetic variants, or structural alterations within the exome, depending on the specific tissue. Although potential discrepancies between blood DNA and tissue DNA exomes may exist, it is worth noting that these differences may not necessarily be significant but could be in a cancer sequencing context[38]. Finally, we did not explore in this study ES performance in additional variants detection in off-target reads, as mitochondrial variants

  1. The paper writing requires more of dedication as the formatting of table/figure arrangement is not appropriate.

We agree. Appropriate modifications have been made.

  1. The overall paper writing, experiment conducted and creativity is excellent

Thank you for your review. We are glad that you find our manuscript excellent.

Reviewer 2 Report

The study aims to provide a structured method for evaluating the performance of exome sequencing strategies as a replacement for targeted sequencing in clinical contexts. The method is based on a set of run-specific and sample-specific sequencing metrics that cover quality and coverage performance on gene panels and OMIM morbid genes. The study evaluated three different exome kits against a myopathy-targeted sequencing method and found that all tested exome kits generated data suitable for clinical diagnosis when achieving 80 million reads per exome. However, significant differences in coverage and PCR duplicates were observed between the kits, indicating that these two criteria are crucial for initial implementation with high-quality assurance. The structured method developed in the study could be useful for molecular diagnostic laboratories in adopting and evaluating exome sequencing kits in a diagnostic context. It also suggests that a similar strategy could be used to implement whole genome sequencing for diagnostic purposes.

The evaluation results are not interesting. The manuscript did not provide valuable information for researchers working on transition from targeted sequencing to exome or genome sequencing for clinical contexts. The authors should declare the net progress of this manuscript compared to their previous work in reference [3]. In addition, I have the following concerns which should be addressed.

Primary comments:

1.    Figure 1 font size is too small to read.

2.    Figure 2 left and right panel is not annotated and described clearly. Especially left panel is not clear for each intersection, i.e., which belons to 0.47, 0.2, 0.04, etc.

3.    Table 1 Source column is not necessary using URLs. It can be replaced by reference.

4.    Table 2 is good, but can the authors highlight the highest performance rows for readers?

5.    Figure 3 four panels should be subtitled and described correctly in figure legend.

6.    In Section 2.4, the author should be more clearer for how to make final selection of a various of strategy based on conditions.

7.    Please check and include related studies: [Samuels, David C., et al. "Finding the lost treasures in exome sequencing data." Trends in Genetics 29.10 (2013): 593-599.] and [Tenedini, E., Bernardis, I., Artusi, V., Artuso, L., Roncaglia, E., Guglielmelli, P., ... & Tagliafico, E. (2014). Targeted cancer exome sequencing reveals recurrent mutations in myeloproliferative neoplasms. Leukemia, 28(5), 1052-1059.]

8.    It is noted that this manuscript needs careful editing by someone with expertise in technical English editing paying particular attention to English grammar, spelling, and sentence structure.

Author Response

Thank you for reviewing our paper

To support the progress that our results will make in identifying the molecular causes of the myopathies studied in the laboratory, compared with previously published work in reference [3], we have added the following sentence in the discussion section : 

“This clinical validation is essential because it assures us of the correct sequencing of the myopathy-genes of our initial panel and of the OMIM genes, which will improve our diagnostic yield of myopathies [3].

Primary comments:

  1.   Figure 1 font size is too small to read.

Thanks for this remark. As requested, Figure 1 was modified to be more readable.

  1.   Figure 2 left and right panel is not annotated and described clearly. Especially left panel is not clear for each intersection, i.e., which belons to 0.47, 0.2, 0.04, etc.

Thanks for this feedback. As requested, Figure 2 was modified to be more readable. Furthermore, a sentence was deleted at the end of the legend.

  1.   Table 1 Source column is not necessary using URLs. It can be replaced by reference.

We agree. Appropriate changes have been made. Table 1 was modified, sources are replaced by references

  1.   Table 2 is good, but can the authors highlight the highest performance rows for readers?

Thanks for this comment. All best performances are now highlighted (bold font).

  1.   Figure 3 four panels should be subtitled and described correctly in figure legend.

Agree. Appropriate changes have been made in the legend.

  1.   In Section 2.4, the author should be more clearer for how to make final selection of a various of strategy based on conditions.

Thank you for this feedback. Appropriate changes have been made to clarify our selection strategy. The new chapter is ”In summary, all evaluated ES kits met clinical diagnostic quality standards based on four run sequencing metrics and seven sample sequencing metrics. Upon reaching 80 million reads, all three kits effectively covered at least 90% of targeted bases at 30X coverage in our GPS and OMIM gene coding regions. The primary factors influencing our ES strategy selection were the PCR duplicate rate, which varies among library kits, and coverage of clinically relevant regions. Notably, the larger target size in the CREV2 kit may present financial constraints for many clinical laboratories when implementing routine diagnostic sequencing.

  1.   Please check and include related studies: [Samuels, David C., et al. "Finding the lost treasures in exome sequencing data." Trends in Genetics 29.10 (2013): 593-599.] and [Tenedini, E., Bernardis, I., Artusi, V., Artuso, L., Roncaglia, E., Guglielmelli, P., ... & Tagliafico, E. (2014). Targeted cancer exome sequencing reveals recurrent mutations in myeloproliferative neoplasms. Leukemia, 28(5), 1052-1059.]

Thanks a lot for these suggestions. We included these related studies that improved the discussion section. 

  1.   It is noted that this manuscript needs careful editing by someone with expertise in technical English editing paying particular attention to English grammar, spelling, and sentence structure.

Following your comment, a review by an English speaker was performed.

Round 2

Reviewer 2 Report

The authors have addressed all my comments. I can accept this version of manuscript.